# The Effect of Downward Social Comparison on Creativity in Organizational Teams, with the Moderation of Narcissism and the Mediation of Negative Affect

**DOI:** 10.3390/bs13080633

**Published:** 2023-07-29

**Authors:** Yuha Yang, Heesun Chae

**Affiliations:** 1Department of Business Administration, Sun Moon University, Asan 31460, Republic of Korea; niceyang@sunmoon.ac.kr; 2School of Business Administration, Pukyong National University, Busan 48513, Republic of Korea

**Keywords:** downward social comparison, creativity, negative affect, narcissism, social comparison theory, hierarchical linear modeling

## Abstract

Employee creativity has become an essential element for the survival and success of contemporary organizations in the fast-changing business environment. The increased importance of team systems in the flood of information has increased the attention given to creativity in social relationships. This study adopts social comparison theory to propose a framework that shows how social comparisons of creative ability between team members influence individual creativity. In particular, this study focuses on the downward social comparison that individuals frequently experience in real team situations. We adopted multi-source field data collected from 130 employees and supervisors working in a manufacturing company in South Korea. The moderated mediation hypotheses were tested using hierarchical linear modeling to address the dependence of employees rated by the same supervisor, given that employees are nested within supervisors. The effect of downward social comparison on creativity, as mediated by negative affect, is positively moderated by narcissism. Specifically, the conditional indirect effects of downward social comparison on creativity through negative affect were significant and negative when narcissism was high but insignificant when it was low. This research provides novel insights for researchers and practitioners by offering a theoretical elaboration of the effects of social comparison processes on creativity and providing unique empirical validation for the model in the context of teams in actual organizations.

## 1. Introduction

Despite the pervasiveness of team-based job designs in contemporary organizations, it remains uncertain and ambiguous how within-team dynamics affect member creativity. The tendency to adopt the team system as a major organizational structure and emphasize creativity as a social process highlights the perspective that social relationships with others in the workplace heavily influence creativity [1,2]. In this regard, team dynamics in organizations have been studied through various theoretical perspectives, such as social identity theory, social categorization theory, social exchange theory, and the attraction–selection–attrition framework [1,2]. The application of these theoretical perspectives has focused on why differences within a team can emerge from specific attitude outcomes such as conflict or cohesion. In addition to the impact of diversity on cognitive attitude formation in groups and group creativity, it is important to review how team members’ perceptions affect diversity dynamics [3]. However, research on this matter is insufficient. Specifically, the social cognitive perspective, which reflects a member’s self-perception when observing others in social interactions, has been neglected, as have external media’s influences on human attitudes and behaviors [3].

The literature on organizational behavior has primarily focused on individual attitudes and performance, and interest has gradually expanded to social processes in organizations. With the increasing complexity of tasks and expertise exceeding the cognitive capability of a single person, contemporary organizations have adopted teams as their major work systems, as team performance can effectively cope with successive organizational changes [4]. Organizations that strive for creative outcomes tend to assign all their capable human resources to the same team (e.g., R&D or task force teams for creative projects). All members of this team are compelled to cooperate or compete with one another. Under such conditions, these individuals compare their abilities, competencies, and performances with those of other members and then relate the comparison results to themselves. Therefore, understanding the social comparison process is critical to gaining insights into the behaviors and performances of employees in contemporary organizations. In this respect, the effects of social cognitive processes within teams on creativity must be investigated using social comparison theory [5,6]. 

One major stream of social comparison studies is the direction of comparison [7]. After the similarity hypothesis proposed by Festinger [5], which asserts that people compare themselves with similar others for self-evaluation, researchers have shifted their focus toward other motivations of social comparison, such as self-improvement and self-enhancement. Such studies have invited additional research into upward and downward comparisons. Downward comparison theory [8] proposes that comparing oneself with someone worse off than oneself may provide self-enhancement. By contrast, an upward comparison to someone better off than oneself is assumed to negatively affect subjective well-being because such a comparison diminishes one’s self-esteem [9,10]. However, subsequent research has challenged this prototypical view. For example, patients with chronic illnesses feel threatened when they make downward comparisons with others in more serious situations [11]. Moreover, Collins [12] proposed that individuals with a positive self-perception benefit from upward comparisons. Hence, upward and downward comparisons have been two important domains of social comparison studies [6,7]. Overall, the literature provides a mixed understanding of the effects of social comparisons, and empirical validation is required to fill the research gap. This study aims to provide empirical evidence on the effect of social comparison through a sophisticated process model.

This study focuses on downward social comparison because managers are interested in employees who outperform or believe they are superior to other team members. We determined how downward comparison perceptions affect employees’ creative performance. While social comparison studies consider various factors, including parental care and attention [13], social rank, group fit, attractiveness [14,15,16], and general ability and opinions [17,18], we used creative ability as the object of comparison to investigate creative performance. In the process by which social comparison affects creativity, emotion is a mediating mechanism that explains in greater depth the connection between social comparison and creative performance. Narcissists, by their nature, are predisposed to believe that other people do not value them sufficiently [19,20]. In this study, we propose that narcissism moderates the effect of downward social comparison on affective reactions. Therefore, this research investigates the relationship between downward social comparison and creativity as mediated by negative affect and considering the moderating role of narcissism. 

This study contributes to the organizational literature in various ways. First, this study enhances the understanding of creativity in group dynamics by investigating creativity from a social comparison perspective in team-based organizations. Second, this study offers a detailed understanding of the relationship between social comparison and creativity by considering emotion as a mediating mechanism. Third, we investigate the moderating role of narcissism in explaining employees’ emotional reactions to social comparison. To date, no empirical investigation has been undertaken in this particular domain, making this study innovative. Therefore, the results of the current study are expected to provide important implications on the creativity literature regarding social dynamics. 

In the following sections, the theoretical literature and formulation of hypotheses are discussed. Next, the data and research method are explained. The following section explains the findings and data analysis based on multi-source data obtained from 130 employees and their managers from a manufacturing company in South Korea. The detailed research model is depicted in Figure 1. In the discussion section, the present findings are compared with the previous literature. Finally, theoretical and practical implications, limitations, and directions for future research are delineated and explained. 

## 2. Theoretical Background and Hypotheses

### 2.1. Social Comparison Theory

Social comparison theory, originally introduced by Festinger in 1954, focuses on the notion that individuals possess an inherent motivation to acquire precise self-assessments [5]. Social comparison is defined as the cognitive process by which individuals evaluate and consider information pertaining to one or more individuals concerning their own self-concept [7]. The theory elucidates the process by which individuals assess their opinions and abilities through comparative analysis with others to reduce uncertainty in these domains and acquire self-definition. 

Social comparison is a cognitive process whereby individuals evaluate themselves in relation to others as a means of self-assessment and measurement. This practice allows individuals to assess their positions and statuses based on their personal standards and emotional perceptions of themselves. As individuals continuously compare themselves with others to estimate their standing on opinions and abilities, social comparison theory has been applied to understand and explain employee attitudes and behaviors under social dynamics. With the development of the literature exploring the direction of social comparison and its effects, studies investigating individual reactions to directional social comparisons (i.e., reactions to downward social comparisons) have drawn the attention of researchers [21]. 

### 2.2. Downward Social Comparison and Negative Affect

Downward social comparison was initially proposed as a means for self-enhancement when facing threats [8]. Early studies examined the positive effect of downward social comparison. They suggested that threatened individuals who need self-enhancement might benefit from downward comparisons to increase their subjective well-being and self-esteem, which, in turn, could lessen their anxiety and produce a pleasant impact [9,20]. Researchers who have delved into individual reactions to downward social comparison in diverse situations have found that this conventional viewpoint may not always be correct. Specifically, they found that comparing oneself favorably to others may not always result in positive sentiments. For instance, people who suffer from chronic diseases have a high risk of feeling intimidated when they compare themselves to patients who are coping with more severe conditions [11]. 

This argument was settled by Buunk et al. [9], who demonstrated that downward comparisons may have both positive and negative effects depending on the situation. Knowing that other individuals are in a worse condition than oneself presents two alternative stories: (a) one’s position is better than that of others, or (b) one’s position may also deteriorate. When the inferiority of others is independent of oneself, individuals may decide to concentrate on the positive aspects of their situation and feel good about it. Conversely, employees focus on unfavorable aspects and experience negative feelings when the poor position of others has a negative influence [22,23]. Buunk and colleagues [24] summarized the consequences of social comparisons on affect, arguing that downward comparisons lead to unpleasant emotions when one is not competing with the target of comparison. 

In organizational teams in which members cooperate and are compensated based on team performance, members collaborate rather than compete with each other. Therefore, downward social comparisons regarding creative ability are related to negative affect because they give employees the impression that their team members are incompetent and lack creativity. Thus, employees who make such comparisons are forced to ensure high team performance. While downward social comparisons may negatively elicit positive affect, this study focuses on negative affect because we expect it to have a stronger influence, as “bad is stronger than good [25]”. Therefore, downward social comparisons between one’s creative ability and that of their team members is expected to elicit negative affect in individuals.

**Hypothesis** **1.***The downward social comparison of creative ability among team members will be positively related to negative affect*.

### 2.3. The Moderating Role of Narcissism

Individuals’ affective reactions to social comparisons differ depending on how they interpret the information derived from such comparisons. An individual’s personality and attitude may influence the affective implications of downward comparisons [9,16,26]. In this study, we aimed to elaborate on the relationship between social comparison and negative affect moderated by narcissistic personality, which is a dimension of personality characterized by grandiose self-perceptions and maladaptive preoccupations with achievements and control [18,19,27]. Narcissistic individuals inherently lack secure self-esteem, so they obsessively seek validation from other people [19,28]. They also lack empathy toward others [28]. Narcissists also have inflated views of themselves and unhealthy preoccupations with success and power [18,19,28]. Consistent with these feelings of grandiosity, narcissistic individuals have a sense of entitlement and expect to receive special privileges. Despite narcissists’ high opinions of themselves, their self-esteem is unstable, and they may react with extreme hostility to perceived threats to their self-concept [29,30]. A low level of creative capacity among team members can lead to reduced team performance or a heightened burden on the focal individual, thereby inducing a sense of threat among individuals in the context of downward social comparisons.

Narcissists’ excessive concentration on the self makes it challenging for them to participate in empathetic behaviors, such as understanding the perspectives of others or feeling sympathy and compassion for others’ suffering [12,31]. As a result of their disregard for others, narcissists lack an understanding of others, struggle to develop and sustain meaningful relationships [32], and often display hostile and aggressive behavior [25,33]. When someone with a narcissistic personality encounters downward social comparisons, their lack of understanding of their peers with low creative ability results in a stronger negative affect. Thus, narcissists experience more unpleasant emotions than other employees when they perceive team members. This is because narcissists are compelled to contribute their skills to aid their peers rather than receive favorable treatment [34]. Therefore, we hypothesize the following:

**Hypothesis** **2.***Narcissism will act as moderator to strengthen the positive relationship between downward social comparison and negative affect*.

### 2.4. Negative Affect and Creativity

Past research on affect demonstrates that it has an important influence on employee creativity. Although some findings show that negative affect may motivate employees to seek existing problems in some situations, in general, negative affect in organizational settings inhibits employees’ creativity. For example, negative affect was found to seriously impede individuals’ cognitive access to extended semantic networks and performance on intuitive assessments of word coherence unless individuals had a strong ability to regulate negative affect [25]. Furthermore, in a meta-analysis, Baas et al. [35] clarified that positive affect promotes cognitive flexibility and creativity, while negative affect, especially in an activated tone, is associated with reduced creativity. The researcher noted that negative affect may promote creativity only in certain cases, such as when negative affect is interpreted as a problem signal [36,37,38]. In line with general findings in creativity studies, we propose the following hypothesis:

**Hypothesis** **3.***Negative affect will be negatively related to creativity*.

Hypotheses 1 and 2 suggest that narcissism moderates the impact of downward social comparison on negative affect. According to Hypothesis 3, negative affect will impact employee creativity. These connections can be seen in our general theoretical model (Figure 1). Although Hypotheses 1–3 can be tested by examining the relevance of individual paths in the model, research shows that testing individual paths is insufficient for proving mediation and moderated mediation effects [39,40]. As a result, we offer a final hypothesis that specifies the moderated mediation effects suggested by our model:

**Hypothesis** **4.***Downward social comparison will be related to creativity via conditional indirect effects, such that its relationship with outcomes will be moderated by narcissism and mediated by negative affect*.

## 3. Methods

### 3.1. Sample and Procedures

Data were collected from employees and their immediate supervisors in a Korean organization that develops and manufactures electric materials. Company managers distributed two sets of surveys to employees and their immediate supervisors. Supervisors of employees were invited to assess the creative performance of members nested within their teams. All participants were provided with a brief statement outlining the study objectives and assuring them of the confidentiality of their responses. When participants finished the questionnaire, they were encouraged to seal the survey envelope and send it straight to the professional researchers. Of the responses we obtained from 170 employees in 32 teams, we identified 130 reliable responses after removing answers with substantial missing information or data that mismatched supervisory ratings.

The employees comprised 86.2% males and had an average age of 38.2 years (SD = 6.15) and an average organizational tenure of 143.9 months (SD = 74.3). Moreover, 1.5% of the respondents had completed only high school, 33.8% had two years of college education, 23.8% had a bachelor’s degree, and 40.9% had a master’s or doctorate degree. 

### 3.2. Measures

All survey items were back-translated according to Brislin’s [41] procedure. Except for the demographic data, responses to all questions were provided on a five-point scale ranging from 1 (strongly disagree) to 5 (strongly agree).

#### 3.2.1. Downward social comparison (DSC) of creative ability

We constructed a four-item measure for the comparison of perceived creative ability between self and coworkers by adopting the creative ability measure of Choi et al. [42]. Downward social comparison was measured with the following referent: “My team members are worse than me at…” (a) “intuitive thinking,” (b) “using their imagination,” (c) “generating new ideas,” and (d) “presenting creative solutions to a given problem.” (α = 0.93).

#### 3.2.2. Narcissism

We adopted four narcissism items from the Dark Triad measure [43] that assess Machiavellianism, psychopathy, and narcissism. Sample items are “I tend to seek prestige or status” and “I tend to expect special favors from others” (α = 0.94).

#### 3.2.3. Negative affect

We used the job-related affect measurement [44] to identify negative affect. We asked the participants to indicate how much they experience specific emotions when working on creative tasks with their coworkers. Each item started with “When I work on creative assignments with my coworkers in my team, I feel...” followed by (a) “depressed,” (b) “dejected,” (c) “hopeless,” (d) “angry,” (e) “frustrated,” (f) “nervous,” (g) “tense,” and (h) “worried.” (α = 0.88) 

#### 3.2.4. Creativity

We constructed a six-item measure to assess employees’ creativity, adapting items from Madjar et al. [45]. Immediate supervisors measured their employees’ creativity (α = 0.89). Example items include (a) “This employee is a good source of highly creative ideas,” (b) “This employee demonstrates originality in his/her work,” (c) “This employee suggests radically new ways for doing his/her work,” (d) “This employee uses previously existing ideas or work in a slightly different fashion,” (e) “This employee is very good at adapting existing ideas,” and (f) “This employee easily modifies previously existing work processes to suit his/her current needs.” 

#### 3.2.5. Control variables

We included employees’ demographic information (age, gender, organizational tenure, and education level) as control variables to rule out alternative explanations, as suggested by prior creativity [38,42]. We measured age in years and organizational tenure in months. Gender was dummy-coded as 1 for males and 0 for females. The highest education attained (1 = high school, 2 = junior college, 3 = undergraduate, 4 = master’s or doctoral degree) was used to determine education level. Table 1 provides the demographic characteristics of participants.

## 4. Results

### 4.1. Confirmatory Factor Analysis (CFA) 

To assess the empirical distinctiveness of the scales employed in this investigation, we conducted confirmatory factor analysis (CFA). We specifically evaluated the fit of the anticipated four-factor measurement model (i.e., downward social comparison, negative affect, narcissism, and creativity) to that of the alternative three- or two-factor measurement models, which aggregated two or three latent variables (e.g., downward social comparison, negative affect and narcissism) to represent a single factor. Table 2 demonstrates that the hypothesized measurement model with four different latent components fits the data well (χ^2^ [202] = 390.322, *p* < 0.01; CFI = 0.909, IFI = 0.910, RMSEA = 0.085). The postulated factor structure also outperformed all alternative measurement models, confirming the distinctiveness and validity of the current scales. 

### 4.2. Correlation Analysis 

The descriptive statistics for all variables included in the research are shown in Table 3. The results showed a positive association between downward social comparison and negative affect (r = 0.30, *p* < 0.001) but a statistically insignificant association between downward social comparison and creativity (r = −0.15, *ns*.). There was a negative correlation between negative affect and creativity ( r= −0.23, *p* < 0.05).

### 4.3. Hypothesis Testing 

We used hierarchical linear modeling (HLM) to address the dependence of employees rated by the same supervisor, given that employees are nested within supervisors [46].

#### 4.3.1. Effect of downward social comparison on negative affect

The findings of the HLM are shown in Table 4. After we controlled for age, gender, education, and organizational tenure, the results revealed a significant relationship between downward social comparison and negative affect (γ = 0.30, *p* < 0.05; Model 2). Therefore, Hypothesis 1 was supported.

#### 4.3.2. Moderating effect of narcissism

Hypothesis 2 suggested that narcissism has a positive moderating effect on the relationship between downward social comparison and negative affect. According to the results in Model 4 of Table 4, the moderation of narcissism was significant (γ = 0.31, *p* < 0.05). 

To further explore the moderating effect, we plotted the simple slopes of two levels of narcissism: one standard deviation above and below the mean [47,48]. As shown in Figure 2, there was a significant positive relationship between downward social comparison and negative affect when narcissism was high (b = 0.564, *p* < 0.001) but an insignificant relationship when narcissism was low (b = −0.034, *ns*.). Thus, Hypothesis 2 was supported.

#### 4.3.3. Negative affect and creativity

We tested the relationship between negative affect and creativity using HLM while controlling the independent variable, the moderator, and their interaction. Model 6 of Table 4 shows that negative affect is negatively related to creativity (γ = −0.14, *p* < 0.05), supporting Hypothesis 3.

#### 4.3.4. Moderated mediation by narcissism

Hypothesis 4 identified narcissism as a moderator in the relationship between downward social comparison and creativity mediated by negative affect. Table 5 shows that the conditional indirect effects of downward social comparison on creativity through negative affect were significant and negative when narcissism was high (conditional indirect effect = −0.07, 95% CI of −0.190 and −0.014) but non-significant when it was low (conditional indirect effect = −0.01, 95% CI of −0.077 and 0.042). Therefore, the present analysis confirmed Hypothesis 4. 

## 5. Discussion

### 5.1. Overall Findings

This study challenges the prevailing assumption that downward social comparison yields positive emotional reactions and outcomes for employees because perceived superiority enhances individuals’ self-esteem. We expand on a previous finding that social comparison may play a role in individual creativity by elaborating on the relationship between downward social comparison and creativity by examining the underlying mechanism accounting for this relationship and its boundary conditions. The present conceptual and empirical analysis demonstrates that downward social comparisons decrease creativity through negative affect, which can be strengthened when an individual is high in narcissism. The following paragraphs outline the major findings discovered in this study. 

First, it was confirmed that downward social comparison is positively related to negative affect, which, in turn, negatively affects creativity. The present findings indicate that downward social comparison negatively influences employee affect, which is consistent with the findings of previous studies [7]. Second, the moderating role of narcissism on the relationship between downward social comparison and negative affect was confirmed. This result implies that this individual trait is an important factor that affects employees’ reactions to social comparisons. Third, this study illustrates the process by which downward social comparison affects creativity. This result confirms the conditional indirect relationship between downward social comparison and creativity through negative affect when narcissism is high. This suggests that even when social comparison results are equivalent among employees, social comparison’s effect on employee creativity via negative affect may vary depending on personal characteristics.

### 5.2. Theoretical Implications

This work makes several important contributions to the literature on creativity. The main contribution is the application of a well-established social psychology theory of social comparison to the literature on organizational creativity. Earlier works frequently utilized a person–situation interactionist perspective for explanation [42,49]. Existing research has also concentrated on intra-individual processes and treated people as independent entities. This line of research has tended to focus on the personal characteristics or social contexts that spur creativity. However, the fact that humans are social animals, as Aristotle noted, has received little attention.

The significance of inter-member dynamics in creativity research has recently attracted attention since modern companies are becoming increasingly reliant on team systems. According to Perry-Smith and Mannucci [50], “the notion that creativity is a social process has increasingly gained prominence.” Numerous attempts have been made [38,42,50] to capture the social aspects of creativity because it is well-known that social relationships with others influence creativity in the work community [51]. Unfortunately, these approaches have failed to address the social cognitive components of creativity, which allow people to learn from others and alter their behavior without directly interacting with them, such as through social exchanges or feedback. 

In addition, research on interpersonal interactions in organizational teams concerning creativity has been restricted to diversity [1,2]. Social comparison processes must be considered when researching creativity in organizational team contexts because people naturally desire to constantly compare themselves to others [4]. Thus, the current study applies the social comparison theory to investigate the effects of interpersonal interactions among team members, a neglected area of research.

Another noteworthy contribution of this study is that it verifies the mediating effect of negative affect in an organizational team setting. This study expands the research scope of the emotion literature and contributes empirically by providing a process model connecting social comparison and creativity. The results show the dominant role of negative affect in the relationship between downward social comparison and creativity. This result suggests that downward social comparison in organizational teams decreases employee creativity by engendering a negative affect.

Finally, this study contributes to the understanding of the effect of downward social comparison on creativity by considering the role of narcissism. In particular, this study provides a new characteristic consideration of downward social comparison studies by combining narcissism and social comparison theory. When employees realize that their colleagues are inferior to them, they experience negative feelings about being solely responsible for the team’s achievements; as a result, their creativity decreases. Employees with high levels of narcissism tend to have an overwhelming need for continual, unjustified adoration and an unreasonably high sense of self-importance [32,34]. Therefore, they consider themselves deserving of favors and preferential treatment. They also grow increasingly frustrated with the fact that their superior self is not regarded, especially by their coworkers, and offered special consideration and instead must share their abilities with their coworkers. 

### 5.3. Practical Implications

This study has several practical implications for organizational managers who want to foster employee creativity within their team systems. The social interactions between team members impact the creativity of employees working in team systems. Negative reactions to downward social comparisons suggest that employees do not want to stand out in a group of weak members but prefer to be a member of a strong team. The perception of one’s membership in a team with low potency increases the negative affect, which lowers employee creativity. Thus, leaders should directly and indirectly encourage employees to believe that they belong to a strong team and imbue them with a sense of pride in their team membership. Leaders and team members should foster mutual respect and honor among team members to promote favorable perceptions of their team. Managers may consider forming teams with members possessing high levels of creative ability so that talented employees can collaborate to produce creative outcomes.

Furthermore, managers should be aware that negative emotions are associated with decreased cognitive motivation, which lowers employees’ interest in creative endeavors. In contrast to individuals assigned to individual tasks, who abandon substantial cognitive efforts when experiencing negative emotions, individuals working in teams interpret negative emotions as a signal to reduce risk-taking behaviors. Thus, individual creativity in work teams would be enhanced by efforts to reduce negative emotions among team members. 

### 5.4. Study Limitations and Future Research

Despite the importance and implications of this study, it has the following limitations. First, this study focuses on the employees of a single Korean manufacturing company, and, as such, the findings have limited generalizability. Future studies are needed to investigate employees from multiple organizations, industries, or countries. Second, the use of cross-sectional data in this study means that only correlational relationships among the study variables can be claimed. Future studies should employ experimental methodologies or collect longitudinal data to establish causal relationships. Third, this study revealed the importance of individual factors when translating social comparison results among team members. Future research might further investigate the role of trait factors in such relationships. For example, the dimensions of consciousness and duty and driving achievements might alter the interpretation of social comparison. Duty has an altruistic orientation, which can alleviate negative perceptions, while the dimension of driving achievements has a competitive and self-centered orientation, meaning the negative effects of comparison can be enhanced. Fourth, this study only examined the effect of DSC, a dimension of the direction of social comparison. Future studies should examine the effects of the two different social comparisons together through the parallel structure of upward social comparison and downward social comparison.

## Figures and Tables

**Figure 1 behavsci-13-00633-f001:**
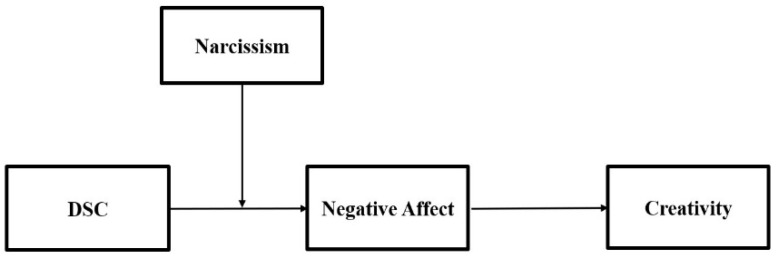
Research model.

**Figure 2 behavsci-13-00633-f002:**
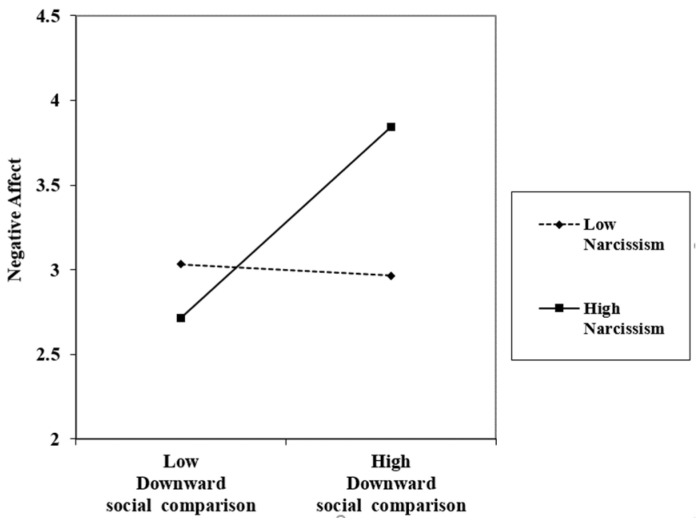
Interaction between downward social comparison and narcissism on negative affect.

**Table 1 behavsci-13-00633-t001:** Descriptive properties of participants.

Spec.		Frequency	Percentage
Gender	Female	18	13.8
	Male	112	86.2
Age	20	12	9.2
	30	68	52.3
	40	45	34.6
	Over 50	5	3.9
Education	High school graduated	2	1.5
	College graduated	44	33.8
	University graduated	31	23.8
	Over graduate school	53	40.9
Tenure	Under 5 years	18	13.8
	5–10 years	29	22.3
	10–15 years	42	32.3
	15–20 years	21	16.2
	Over 20 years	20	15.4

**Table 2 behavsci-13-00633-t002:** Confirmatory factor analysis.

Model	No. of Factors ^a^	χ^2^	df	Δχ^2^	RMSEA	CFI	IFI
Baseline model	4 factors: DSC, NA, NAR, CRE	390.322	202		0.085	0.909	0.910
Model 1	3 factors: (DSC + NA), NAR, CRE	659.200	205	268.878 **	0.131	0.780	0.783
Model 2	2 factors: (DSC + NA + NAR), CRE	1005.551	207	615.229 **	0.173	0.613	0.617

Note: **: *p* < 0.01, ^a^ DSC = Downward social comparison; NA = Negative affect; NAR = Narcissism, CRE = Creativity, RMSEA = Root mean square error of approximation, CFI = Comparative fit index, IFI = Incremental fit index.

**Table 3 behavsci-13-00633-t003:** Descriptive statistics and correlation analysis between variables.

Variable	Mean	S. D	1	2	3	4	5	6	7	8
1. Age	37.12	6.15								
2. Gender	0.42	0.35	0.16							
3. Tenure	2.82	74.26	0.83 ***	0.18 *						
4. Education	7.66	1.10	−0.29 **	−0.20 *	−0.38 ***					
5. DSC	4.80	0.65	0.32 ***	0.08	0.22 *	−0.19 *	<0.93>			
6. Narcissism	4.00	0.68	−0.70	−0.13	−0.07	0.06	0.08	<0.94>		
7. Negative affect	4.80	0.71	0.07	0.06	0.01	−0.21 *	0.30 ***	0.10	<0.88>	
8. Creativity	4.80	0.54	0.10	0.11	0.07	0.04	−0.15	0.04	−0.23 *	<0.89>

N = 130, *: *p* < 0.05, **: *p* < 0.01, ***: *p* < 0.001. Reliabilities are on the diagonal in parentheses.

**Table 4 behavsci-13-00633-t004:** Results of HLM analysis.

Variable	Negative Affect				Creativity		
	Model 1	Model 2	Model 3	Model 4	Model 5	Model 6	Model 7
Constant	2.10 **	2.51 **	2.45 **	2.23 **	2.29 **	2.28 **	2.26 **
Age	0.03	0.01	0.01	0.01	0.02	0.02	0.02 *
Gender	0.05	0.04	0.07	0.15	0.22 *	0.21	0.00
Tenure	0.00	0.00	0.00	0.00	0.00	0.00	0.00
Education	−0.15 *	−0.13 *	−0.13 *	−0.11 *	0.03	0.01	-10
Downward social comparison (DSC)		0.30 *	0.28 *	0.25 *		−0.14 *	
Narcissism (NAR)			0.10	0.15			
DSC X NAR				0.31 *			
Negative affect							−0.11 *
Pseudo R square	0.48	0.51	0.51	0.53	0.01	0.05	0.08

Note: N = 130, *: *p* < 0.05, **: *p* < 0.01.

**Table 5 behavsci-13-00633-t005:** Bootstrapped moderated mediation result.

Independent Variable	Mediator	Dependent Variable	Moderator	Moderator Level	Conditional Indirect Effect	Bootstrapping Bias-Corrected 95% Confidence Interval	
						Lower	Upper
Downward Social Comparison	Negative Affect	Creativity	Narcissism	Low (Mean − 1 SD)	−0.009	−0.077	0.042
				**High (Mean + 1 SD)**	**−0.073**	**−0.190**	**−0.014**

Note: Bootstrap sample size = 1000. Coefficients in bold indicate significant mediation.

## Data Availability

Not applicable.

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
