# Peer review of "The Effect of Downward Social Comparison on Creativity in Organizational Teams, with the Moderation of Narcissism and the Mediation of Negative Affect"

_behavsci, 2023, doi:10.3390/bs13080633_

Round 1

Reviewer 1 Report

Dear Authors,

The presented research topic is current and extremely important. The issue of employee creativity is part of the concept of entrepreneurial management. Many researchers have proven the positive impact of entrepreneurial orientation on the results achieved by organizations. The research presented in the article allows us to understand how employee teams should be built to strengthen their creativity. I think that the article raises important theoretical and practical issues. The presented results may be an impulse for further research on employee creativity and teamwork. I think that the purpose of the research has been achieved.

Congratulations on undertaking such an interesting research topic, and I wish you success.

Reviewer 2 Report

The topic of the paper is pertinent and current. The keywords of the topic and its relationships are well chosen.

The abstract has all the fundamental parts to the understanding of the study.

The introduction has the fundamental parts to understand the pertinence and objectives of the research. It is suggested to the authors the formulation of a research question.

The literature review is focused excellently on the keywords of the topic and which are the object of study.

The methodology and statistical methods followed are appropriate. 

In section 5, the authors are suggested to compare the results with those of the literature review.

It is suggested to the authors to make a section with the conclusions, in which it should have the following topics: Main results; Limitations; future investigations; theoretical and practical implications.

Reviewer 3 Report

Dear author(s)

Thank you for giving me the opportunity to read your paper. I have a few suggestions that you may consider as you develop the paper further. please refer to my comment. it needs a Major revision and you should make all the comments with the highlight in the article.

Best wishes

Why you did this research i.e. what is the practical gap?

The introduction should clearly illustrate (1) what we know (the key theoretical perspectives and empirical findings) and what do we not know (major, unaddressed puzzle, controversy, or paradox does the study addresses, or why it needs to be addressed and why this matters). And, (2) what will we learn from the study and how does the study fundamentally change, challenge, or advance scholars’ understanding. Much sharper problematization is required so that the introduction draws the reader into the paper. The introduction therefore needs to do a better job in setting the stage for the articulation of the theoretical contributions of the study. At the end of the introduction, we should have a clear idea of what the paper is about (i.e. its motivation, the gap in understanding that the paper is trying to address and summary of theoretical contributions).

Theoretical Framework: it is better to clearly dived it to subs-sections .

Theoretical literature has not been considered and reviewed. It’s better to observe the connection between the contents. Try to explain everything except the topics in order to establish the necessary coherence.

Theoretical Development: The literature review must engage in the constructs of your analytical framing in a meaningful way. The literature review section could be improved by being more analytical. In other words, building on the existing literature to highlight what is missing and what is yet to be done and in so doing outline the theoretical puzzles or debates to which this work contributes. I have concerns related to theoretical development, and note the need for a more rigorous critique of the literature to help deepen the theoretical underpinnings of the study.

Discussion: please first state the result then discusses it or justify it. please restructure this section. 

Conclusion: First start with results then implications, contributions, and future research, do not mix them.

What are the limitations of this research and how can it be solved by other researchers?

Using the following reference could be beneficial as these add more evidence to the literature review section:

(2023). The effect of team performance on the internationalization of Digital Startups: the mediating role of entrepreneurship. International Journal of Human Capital in Urban Management, 8(1), 17-30.

Best of luck with the further development of the paper.

It needs minor revision.

Round 2

Reviewer 2 Report

The changes made in the paper are significant and their result ensures a coherent line of research between the topic, abstract, keywords, results, conclusions and lines of future research.

Reviewer 3 Report

Dear author(s)

Hope you are doing well. According to the review of this article, the corrections have been made.

Good luck

Hi Dear Editor

Hope you are fine.

Thank you for giving me the opportunity to again review this paper. According to the review of this article, the corrections have been made by the authors.

Best regards

Mehdi